# Modern Synergetic Neural Network for Synthetic Aperture Radar Target Recognition

**DOI:** 10.3390/s23052820

**Published:** 2023-03-04

**Authors:** Zihao Wang, Haifeng Li, Lin Ma

**Affiliations:** Faculty of Computing, Harbin Institute of Technology, No. 92, Xidazhi Street, Nangang District, Harbin 150001, China

**Keywords:** SAR target recognition, feature extraction, fusion model, synergetic neural network, autoencoder, prototype learning

## Abstract

Feature extraction is an important process for the automatic recognition of synthetic aperture radar targets, but the rising complexity of the recognition network means that the features are abstractly implied in the network parameters and the performances are difficult to attribute. We propose the modern synergetic neural network (MSNN), which transforms the feature extraction process into the prototype self-learning process by the deep fusion of an autoencoder (AE) and a synergetic neural network. We prove that nonlinear AEs (e.g., stacked and convolutional AE) with ReLU activation functions reach the global minimum when their weights can be divided into tuples of M-P inverses. Therefore, MSNN can use the AE training process as a novel and effective nonlinear prototypes self-learning module. In addition, MSNN improves learning efficiency and performance stability by making the codes spontaneously converge to one-hots with the dynamics of Synergetics instead of loss function manipulation. Experiments on the MSTAR dataset show that MSNN achieves state-of-the-art recognition accuracy. The feature visualization results show that the excellent performance of MSNN stems from the prototype learning to capture features that are not covered in the dataset. These representative prototypes ensure the accurate recognition of new samples.

## 1. Introduction

With the development of synthetic aperture radar (SAR) technology, the explosive growth of SAR images has presented new challenges for highly accurate target recognition. Compared with manual feature extraction methods [1,2,3,4], deep learning techniques represented by the autoencoder (AE) enable the automatic extraction of target features [5], achieving a better performance with improvements in efficiency, and are therefore widely used in SAR image recognition tasks. With the development of deep learning, the basis model evolved from stacked AE [6,7,8] and sparse AE [9,10,11] to convolutional AE [12,13,14]. The evolution of the model enhances its accuracy. However, with the rising complexity of the network structure, the extracted features are implicitly and increasingly abstracted inside the network, making it difficult to determine whether the features are effectively extracted and to explain the excellent performance of the network.

During the application of AE to SAR, a fusion model of AE and the synergetic neural network (SNN) is proposed [15]. AE-extracted features are transmitted to SNN to improve the robustness of classification. However, many outstanding SNN characteristics are not fully utilized. SNN handles non-binary data [16,17,18] with an excellent theoretical performance [19], and all its attractors in the dynamic system correspond to valid memories. These characteristics mean that the SNN’s classification is actually an association between the features and the extracted prototypes [20,21,22]. SNN’s simulation of associative memory achieves remarkable results in image retrieval [23], face recognition [24], and semantic role annotation [25,26]. Its synergetics-based dynamics [27] can improve the recognition performance of the network and optimize the pattern extraction process by constructing an attractor–prototype correspondence. Therefore, significant breakthroughs can be obtained from the SNN-optimized AE feature extraction for recognition performance enhancement.

In this paper, we investigate the fusion method of AE and SNN and propose the modern SNN (MSNN) model. More than a decade before the formal introduction of AE, researchers showed that the weight matrices of the encoder and the decoder from single-layer linear AE are mutual M-P inverse if the loss function reaches a stationary point, including one global minimum and multiple saddle points [28], which are precisely in line with the requirements of the SNN prototype and adjoint matrix. Inspired by this study, we extended this conclusion to multilayer nonlinear AE and prove that its weight matrices can be regarded as multiple sets of M-P inverses during convergence. Next, we proposed MSNN. We treated the encoder and decoder as the generalized adjoint and prototype features of SNN and used the pattern extraction of the AE to realize nonlinear prototype self-learning. The synergetic dynamics allow for the codes of MSNN to converge to one-hot in a more stable and controllable manner than sparse or adversarial AEs. The binarized codes can be efficiently visualized through associative memory. Experimental results on the MSTAR dataset [29] show that MSNN outperforms sparse AEs, marginally outperforms convolutional AE without coding restrictions, and effectively visualizes the extracted features. Visualization results show that the self-learned prototypes reconstruct the sample features not contained in the dataset, and the reconstructed images are noise-free, which proves that the MSNN can successfully learn unobserved representative features for target recognition.

The objective of this paper is to propose an SNN-AE fusion model MSNN to learn nonlinear prototypes as representative features of SAR images to improve recognition accuracy. The contributions of this paper are (1) proposing the fusion model of SNN and AE with complementary strengths, (2) proposing the prototype self-learning method to enhance feature representativeness and visualization, and (3) attributing the recognition performance of deep learning to the effective learning of unobserved features.

This paper is structured as follows. Section 2 provides the background of the study. Section 3 introduces the fusion model MSNN. The first two sections provide the theoretical basis. Section 3 describes MSNN’s working process and learning method. Section 4 demonstrates the experiment configurations, results, and analysis. Section 5 summarizes the whole paper and offers the limitations, unexplored perspectives, and future directions of our study. This paper proposes a novel artificial neural network model to solve the feature recognition problem of SAR images, which may be of great interest to radar and artificial intelligence scientists, researchers, and trainees and may provide assistance in their research.

## 2. Background

### 2.1. SNN Overview

Synergetics use dynamic systems to study the dynamic process of multiple subsystems from disordered states to ordered states [27]. SNN, as an application of synergetics for associative memory tasks in computer science, considers the dynamic query pattern x as a disordered state and the static memory v as an ordered state and uses the dynamical system to form the variation from the query pattern to a stored memory in order to simulate the human associative memory process. For the query pattern x and the matrix of the static prototypes $V=[v_{1},\ldots ,v_{N}]$ representing memories, the updated formula of SNN [19] with the default hyperparameter setting is as follows:(1)$$\xi =V^{+}x$$
(2)$$\xi^{\mathrm{new}}=\mathrm{Syn}\left(\xi \right)=\gamma \left(\frac{\xi^{3}+\xi }{{2∥\xi ∥}_{2}^{2}}+\left(\frac{1}{\gamma }-1\right)\xi \right)$$
(3)$$x^{\mathrm{new}}=V\xi^{\mathrm{new}}$$
where $V^{+}$ is the Moore–Penrose inverse of *V* [30,31], Syn can be interpreted as a synergetics-based activation function, and γ is the learning rate. Equation (Equation 3) is derived from the enslaving principle of synergetics, representing the dynamics of v, which are dominated by ξ. The variation in x is derived through the dynamic update to the ξ. Equation (Equation 1) transforms x to ξ, and Equation (Equation 2) updates ξ. The network repeatedly takes $x^{\mathrm{new}}$ as the new input until it no longer changes, at which point the network converges. As shown in Figure 1, SNN converges to three kinds of stationary points: the target stable point, the saddle point, and the local maxima point. The target stable point is ξ, which reaches the positive or negative one-hot encoding, which is the general case. The network outputs a single prototype ±v, reflecting the association from x to v. The saddle point is ξ, reaching multiple identical non-zero values encodings that stem from multiple equal extremes in the initial value of ξ. The local maximum point is that all elements of ξ are initialized with 0; the network cannot be updated due to the division by 0 steps. SNN restricts the independence of all v and their number to less than its dimension, such that $V^{+}V$ is the identity matrix. Two consecutive iterations of the network were examined and substituted (Equation 3) into (Equation 1), $\xi =V^{+}x^{\mathrm{new}}=\xi^{\mathrm{new}}$. Thus, the update formula can be simplified to the repeated application of Syn to ξ.

### 2.2. Relation of Linear AE Weights

For a single-layer, linear, unbiased, activation-free AE, the single global minimum of the loss function corresponds to the encoder–decoder weights, which are mutually M-P inverse. Suppose the covariance matrix of the trainset is full rank and the loss function contains a global minimum and multiple saddle points; when the global minimum is reached, the network output is an orthogonal projection to the span of eigenvectors of the covariance matrix, and the network is equivalent to an orthogonal projection matrix [28]. By Moore’s definition of the M-P inverse, the weight matrix of the encoder and the decoder are mutually M-P inverse [30], the decoder matrix is composed of eigenvectors, and the codes are the linear combination weights.

## 3. MSNN

We propose an AE-SNN deep fusion model MSNN by proving that the global minimum converged stacked or convolutional AE weights can be divided into tuples of M-P inverses and are suitable for the generalized prototype and adjoint matrices. MSNN replaces the adjoint and prototype matrices with the AE’s multilayer structure, so the prototype learning method is upgraded to nonlinear.

### 3.1. Stacked AE for Prototype Learning

The weights of the global minimum-converged AE can be divided into tuples of M-P inverses. The working process of the nonlinear AE layer is
(4)$$z=W^{T}x+b$$
(5)y=σ(z)
Let $x^{\prime }={\left(x,1\right)}^{T}$, construct
(6)$$W^{\prime }=\left[\begin{array}{cc} W & 0\\ b{}^{T} & 1 \end{array}\right]$$
Thus,
(7)$$z^{\prime }={\left(z,1\right)}^{T}={\left(W^{\prime }\right)}^{T}x^{\prime }$$
The activation function σ scales the elements of z to a specific interval, which is equivalent to applying a $z^{\prime }$-related scaling factor to the corresponding column of $W^{\prime }$. Let the dimension of x be *N*; for 1≤n≤N, we construct the scaling parameter
(8)$$i_{n}=\left\{\begin{array}{c} \frac{y_{n}}{z_{n}},z_{n}\neq 0\\ 0,z_{n}=0 \end{array}\right.$$
Thus, all kinds of σ except Sigmoid can be substituted by *i* (0 cannot be scaled to 0.5 by *i*). Construct $i={\left(i_{1},\ldots ,i_{N},1\right)}^{T}$,
(9)$$y^{\prime }={\left(y,1\right)}^{T}={\left(i^{T}\circ W^{\prime }\right)}^{T}x^{\prime }$$
where “∘” is the Hadamard product. A different $x^{\prime }$ relates to different i, which yields different equivalent matrices. All possible constructions of the equivalent matrices substitute multiple linear AEs for the nonlinear AE. For instance, when σ is ReLU, $i_{n}=1$ for $z_{n}>0$ and $i_{n}=0$ for $z_{n}\leq 0$. For different *x*, each *i* has two possible values, so i has a total of $2^{N}$ possible values. When σ is tanh, $i_{n}$ has a different value for a different $z_{n}$. Therefore, it is almost impossible to find different *x* corresponding to the same *i*. The possible values of i will depend on the number of samples. The linear AEs have an identical loss function to the nonlinear AE, and the added dimension of x and y does not generate additional gradients during the error backpropagation process, so these linear AEs converge to the global minima when the nonlinear AE reach the global minima. According to Section 2.2, the encoder and decoder weights are tuples of mutual M-P inverse.

The above conclusion also applies to stacked AE. The stacked AE can be regarded as applying multiple equivalent matrices $i^{T}\circ W^{\prime }$ to $x^{\prime }$. The product of these matrices can be regarded as the single equivalent matrix, so the weights of the global minimum converged stacked AE weights can also be divided into tuples of M-P inverses.

### 3.2. Convolutional AE for Prototype Learning

The convolutional and transposed convolutional layer can be transformed into equivalent fully connected layers, so the above conclusion can be generalized to convolutional AE. By inputting a feature map *X* of size (H,W) into a convolutional layer with kernel size *K* and stride 1, the following output can be obtained:(10)$$z_{ij}=\sum_{i=1}^{H-K+1}\sum_{j=1}^{W-K+1}\left(b_{ij}+\sum_{k_{1}=1}^{K}\sum_{k_{2}=1}^{K}w_{k_{1}k_{2}}x_{\left(i+k_{1}\right)\left(j+k_{2}\right)}\right)$$
This process can be implemented by constructing an equivalent matrix. Construct $w_{k}={\left(w_{k1},\ldots ,w_{kK},0,\ldots ,0\right)}^{T},$
k=1,2,…,K−1 of dimension W and $w_{K}={\left(w_{K1},w_{K2},\ldots ,w_{KK}\right)}^{T}$ of dimension *K*, and put them together as $w={\left(w_{1},\ldots ,w_{K}\right)}^{T}$. Stack w along the principle diagonal to obtain
(11)$$A=\left[\begin{array}{ccc} w & \ldots & 0\\ \vdots & \ddots & \vdots \\ 0 & \ldots & w \end{array}\right]$$
of size (KW,W−K+1). By stacking A along the principle diagonal and filling the rest with zero matrix *O* of size (W,W−K+1), we obtain
(12)$$W=\left[\begin{array}{cccc} A & O & \ldots & O\\ O & A & \ldots & \vdots \\ \vdots & \vdots & \ddots & O\\ O & O & \ldots & A \end{array}\right]$$
of size (HW+1,(W−K+1)(H−K+1)). Construct $x^{\prime }$ with elements of *X* and 1 as the last dimension; the convolution is equivalent to
(13)$$z^{\prime }={\left(z,1\right)}^{T}={\left(W^{\prime }\right)}^{T}x^{\prime }$$
Figure 2 visualizes the above process. The equivalent matrix of the transposed convolutional layer can be obtained by removing the transpose on $W^{\prime }$.

### 3.3. MSNN Model

The structure of MSNN is shown in Figure 3. For the input x, decoder output $x^{\mathrm{new}}$, code h, and classifier output y, the working process of network modules is as follows:(14)h=Encoder(x)
(15)$$h_{i}^{\mathrm{new}}={\mathrm{Syn}}^{N}\left(h_{i}\right),i=1,2,\ldots ,M$$
(16)$$x^{\mathrm{new}}=\mathrm{Decoder}\left(h^{\mathrm{new}}\right)$$
(17)y=Classifier(h)
We use convolutional layers as the main body of the coders and add a fully connected layer adjacent to h to integrate or recover features of the feature maps. h is equally partitioned into *M* subcodes $h_{1}$ to $h_{M}$ and input in *M* SNNs for *N* iterations.

We introduce multiple SNNs to ensure the reconstruction quality of the decoder by extending the possible values of $h^{\mathrm{new}}$. The convergence target of SNN is one-hot, i.e., one code equals one, and other codes equal zero. Let the dimension of the code be *H*. When a single SNN is used to receive h, there are only *H* different positions for the value one, and thus the total number of possible codes is also *H*. The limited code number means that many samples correspond to one code; however, only one code can be reconstructed to one output. Such an output is usually composed of the common features among those samples, which makes it difficult to cover the representative information. Features lacking representation will damage both the visualization and the recognition performances. When multiple SNNs receive h, each SNN has H/M positions for the value one. Therefore, the total number of $h^{\mathrm{new}}$ obtained by *M* SNNs increases to ${\left(H/M\right)}^{M}$. The total number of codes is exponential to *M*, so the one-to-one correspondence between the input and $h^{\mathrm{new}}$ can be achieved by expanding the value of *M*.

The previous sections only prove that the stacked AE and convolutional AE weights are tuples of M-P inverses when the global minimum is reached, yet the actual convergence state requires additional verification. From Section 2.2, the weights of the decoder consist of the eigenvectors of the covariance matrix. Since the covariance matrix is symmetric and full rank, the eigenvectors are orthogonal to each other. From the property of the M-P inverse, if a matrix *V* has linearly independent columns, $VV^{+}$ is the identity matrix. Extending this conclusion, the product of the decoder’s equivalence matrix and the encoder’s equivalence matrix is also the identity matrix when the stacked or convolutional AE reaches the global minimum. Therefore, the code remains unchanged after recoding:(18)$$h^{\mathrm{new}}={\mathrm{Syn}}^{N}\left(\mathrm{Encoder}\left(\mathrm{Decoder}\left(h^{\mathrm{new}}\right)\right)\right)={\mathrm{Syn}}^{N}\left(\mathrm{Encoder}\left(x^{\mathrm{new}}\right)\right)$$
We introduce the verification term from the equation above to quantify the convergence state.
(19)$$P=\mathrm{MSE}\left(h^{\mathrm{new}},{\mathrm{Syn}}^{N}\left(\mathrm{Encoder}\left(x^{\mathrm{new}}\right)\right)\right)$$
MSE is the mean squared error. The closer the *P* is to zero, the better the parameters are trained. Note that the loss function of the coder is still $MSE(x,x^{\mathrm{new}})$. *P* is only for convergence verification; it is not involved in any parameter tuning process.

For error backpropagation, Syn repeatedly imposes a polynomial function onto the input, which may lead to the gradient exploding or vanishing. The gradient problem is so severe that conventional means such as gradient clipping can barely circumvent the non-convergence. Therefore, we used the gradient bypass technique [32,33] from the incomputable partial solution (Figure 3 red arrow). This technique passes the gradient of the certain network layer and outputs directly to the input during backpropagation in order to circumvent the inappropriate activation functions causing the gradient to explode or vanish and even the gradient intransmissible caused by discontinuity.

The visualization of extracted features can be easily achieved using the recall of MSNN. Although the MSNN prototypes can be derived theoretically by calculating the decoder’s equivalent matrices, the number of equivalence matrices is enormous and difficult to filter. From Section 3.1, the number of equivalent matrices depends on *i*. *i* is exponential to the number of neurons with ReLU activation, and even infinite with other activations, so the calculation of equivalence matrices is time-consuming or even non-traversable. In addition, most of these matrices will not occur in the application due to the limited number of training samples and the intrinsic connection among codes; therefore, it is inefficient and difficult to discover the core prototypes by this approach. Alternatively, we derived the visualization results by inputting a specific code to the decoder. For example, one-hot codes were input to observe individual prototypes, and clustering centers of inner-class codes were input to observe their common features.

## 4. Experiments

### 4.1. MSTAR Dataset Configuration

We designed experiments using SAR images of 10 different military carriers from the MSTAR dataset [29]. The carriers included 2S1, BMP2, BRDM2, BTR60, BTR70, D7, T62, T72, ZIL131, and ZSU234. X-band SAR generates these images with a resolution of 0.3 m × 0.3 m. The images acquired from 17° and 15° were used as the training and test sets, respectively, and there were 2747 and 2426 samples with a near-uniform distribution; details are shown in Table 1. Since the images of different classes varied in size, although the pivotal features were located in the center of the image, we normalized all image sizes to 256 × 256 and used the 64 × 64 pixels in the center as the network input. To minimize the differences in data distribution between the training and test sets, we applied affine transformations to train samples, including random rotation (−15°,15°) and random shear (−15°,15°) in the horizontal and vertical directions. The batch size was 256.

### 4.2. MSNN Configuration

The encoder of the MSNN contained 4 convolutional layers and 1 fully connected layer with the channel configuration 32–64–64–64. This channel configuration ensures that the code dimension is not larger than the input and avoids overparameterization of AE. The kernel size was 4, the stride was 1, and the padding was 1. Due to the limitation of the transpose convolutional layer, having an identical configuration to the convolutional layer is not enough to obtain an output with an equal size to the input. Therefore, we used the common structure to ensure identical data size. Each convolutional layer was followed by a dropout of $D_{conv}$. The neuron number of the fully connected layer equals its input dimension. The encoder’s output h was divided into *M* parts, input to the SNN, and iterated *I* times. The hyperparameter γ = 1. SNN works properly when γ is no greater than 1 [19], so this configuration allows for the fastest convergence. To improve the efficiency of the SNN and avoid the error from the all-zero input, we normalized the input before each iteration and directly output the all-zero code. The decoder and encoder were configured symmetrically. All network layers use the ReLU activation function except the activation-free output layer of the decoder. The classifier inputs h and outputs the category label prediction. To show that the feature extraction of SNN substantially reduces the recognition difficulty, the classifier contains only 1 fully connected layer with a pre-layer dropout of $D_{class}$. The weights of all network layers were initialized using Xavier uniform distribution to match the ReLU activation. The MSE loss function was used for the feature reconstruction task, and the cross-entropy loss function was used for the feature recognition task. The network iterated 120 epochs. Since the feature reconstruction task is simpler than recognition, the recognition task is ignored by MSNN when the training ratio of the two is 1:1. We determined the training ratio of reconstruction to recognition to be 1:10 after pre-experiments. The optimizer was AdamW [34] for faster training, and the learning rate was adjusted using the OneCycle [35] approach to escape the local minimum with the defaulted maximum 1 × 10${}^{-3}$. Gradients were cropped to a defaulted upper limit of 1 under Euclidean length to avoid accidental network divergence. The optimal value of the letter-denoted hyperparameters is discussed in the next section.

### 4.3. Recognition Results Analysis

The feature recognition results of MSNN were optimized with a hyperparameter grid search with ablation test, as shown in Table 2, and compared with other methods in Table 3, and the confusion matrix is shown in Table 4. The validation term *P* reduced from the order of $1\times 10^{-7}$ to $1\times 10^{-8}$, which satisfies the application premise. The optimal result of MSNN was obtained at epoch 71 by the early stopping technique. From Table 2, SNN improves the AE recognition performance for all dropout rate configurations. A larger dropout rate needs to correspond to a larger code segment number with fewer iterations to obtain a better performance. Since deep learning is highly dependent on the amount of data and is sensitive to noise, data augmentation and denoising are widely used in these methods to reduce the application difficulty and obtain an accuracy close to 100%. To maintain a uniform task difficulty compared to other studies, we only compared the results with studies that did not use data manipulation. As a result, MSNN outperformed other methods and achieved results with data augmentation. In addition, the recognition result of MSNN achieved 100% recognition accuracy in five classes, which is also a new state-of-the-art result.

For the MSNN training speed, the network contains 4 convolutional layers, 4 transposed convolutional layers, 1 fully connected layer, and a set of SNN with 16 iterations. SNNs work in parallel and can be approximated as 16 network layers. Therefore, the training speed of the MSNN is roughly equivalent to that of a 25-layer deep network. We used an RTX 3070 graphics card and CUDA 11.6 + pytorch 1.12 environment to compare the time required for one epoch of MSNN with several other methods, and the results are shown in Table 5. The training speed of MSNN is moderate. Although the recurrent architecture prolongs the training time to some extent, it still takes less time than DeepMemory, which has more layers to train.

### 4.4. Prototype Analysis

We first evaluated the effectiveness of the prototypes by counting the average percentage of one-hot codes and zero vectors of subcodes. One-hot represents one prototype that learns the feature. The higher the percentage, the better the effectiveness. The total percentage of the one-hot and zero vectors can characterize the convergence of each SNN. The higher the percentage, the more networks reach convergence. The statistics of the training and test sets are shown in Figure 4. Almost all subcodes are one-hot or zero-vector forms, which means that the MSNN prototype can learn effectively. The proportions of both codes in the train and test sets are similar, reflecting the consistent representation of prototypes. The optimal recognition accuracy is achieved at a low one-hot ratio, implying that reducing the sparsity of the code can improve the recognition performance.

We then visualized the prototype patterns obtained by the MSNN self-learning. Since the prototypes of each SNN capture partial sample features and their combination is nonlinear, the visualization of single prototypes is vague and indistinguishable. Therefore, we clustered similar codes of inner-class training samples by K-means using *K* = 8 and visualized the binarized clustering centers. The results are shown in Figure 5. We also listed the most similar images from the train-set to judge whether the visualization results are merely the simple reconstruction of the input samples. The comparisons show that some visualizations significantly differ from the most similar samples, indicating that the prototypes can learn key angle features that are not covered in the dataset.

## 5. Conclusions

In this paper, we proposed the use of an AE-SNN fusion model MSNN for the feature recognition task of SAR images. AE solves SNN’s lack of prototype nonlinear learning, and SNN improves the efficiency and stability of AE coding regularization and offers a simple and effective feature visualization approach. Experiments on the unpreprocessed MSTAR dataset showed that MSNN obtained the optimal feature recognition performance, and the feature visualization results show that these excellent results originate from the network’s effective prototype learning, which spontaneously captures representative features that are not covered in the training set. It is worth noting that the feature extraction and visualization of MSNN are based on the premise that the critical information is located at the center of the SAR image. For data with a more complex distribution, the features learned by MSNN will be more abstract, so more sophisticated visualization methods need to be developed to ensure performance. In addition, the recognition performance of MSNN on partially missing, mislabeled, and new category samples also needs more studies to achieve a broader application. Target recognition by the fusion of SAR images and high-resolution range profile (HRRP) is a research hotspot. We plan to apply MSNN to SAR-HRRP feature matching and feature fusion to further improve the recognition accuracy.

## Figures and Tables

**Figure 1 sensors-23-02820-f001:**
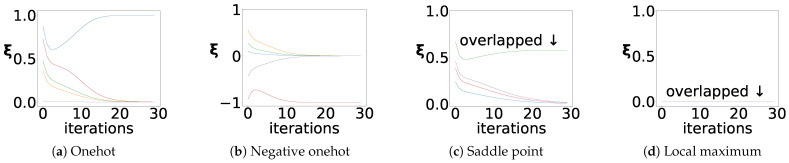
SNN’s convergence to stationary points. Curves with different colors represent different order parameters. In the figure, (**a**,**b**) converge to the target stable point, and the positive or negative one-hot, i.e., one-order parameter converges to ±1, while others converge to 0; (**c**) converges to the saddle point that stems from multiple identical extreme values in ξ; (**d**) converges to the local maximum point that stems from the zero-vector initialization of ξ. Note that this chart is only for representation. The divide-by-0 error terminates the iteration in the update formula.

**Figure 2 sensors-23-02820-f002:**
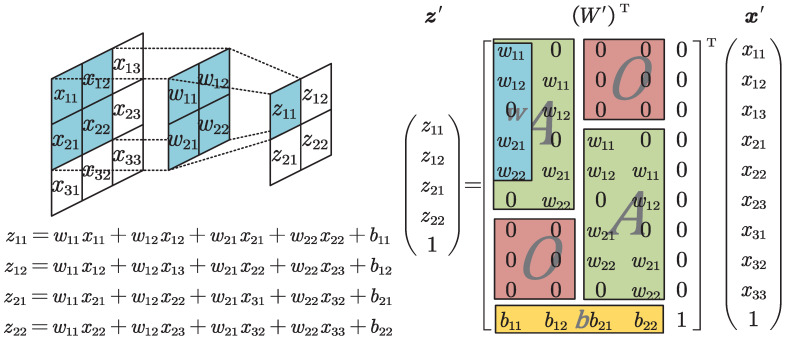
The equivalent matrix construction for the convolutional layer. The left side shows how the convolutional layer works, and the output is summed by bitwise multiplication between the convolution kernel and the input that it covers. The right side shows the construction method of the equivalent matrix $W^{\prime }$. The elements of z have the same operation procedure as the left side.

**Figure 3 sensors-23-02820-f003:**
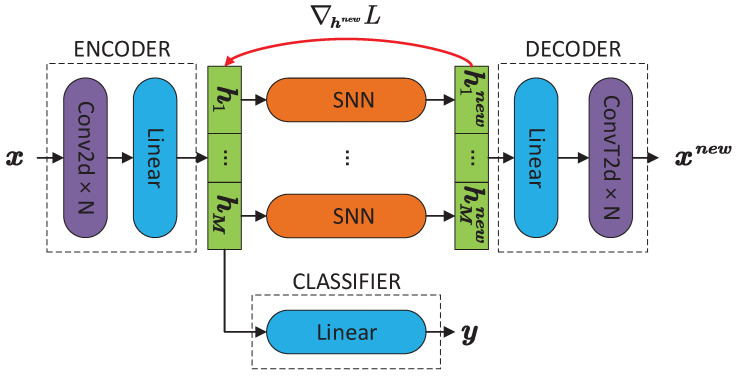
The network structure of MSNN. The encoder inputs x and outputs code h. h is divided into *M* subcodes before inputting to *M* SNNs and obtaining the updated code $h^{\mathrm{new}}$. The decoder inputs $h^{\mathrm{new}}$ and outputs $x^{\mathrm{new}}$. Meanwhile, h is input to the classifier to obtain the label y. Parameters are trained using the error backpropagation. The loss function of the coders is $MSE(x,x^{\mathrm{new}})$ (MSE is mean squared error), and the loss function of the classifier is CrossEntropy(y,label). The gradient of $h^{\mathrm{new}}$ from error backpropagation bypasses SNN and directly transmits to h. The training of the coders is monitored by the value of *P*. *P* approaching zero means good training.

**Figure 4 sensors-23-02820-f004:**
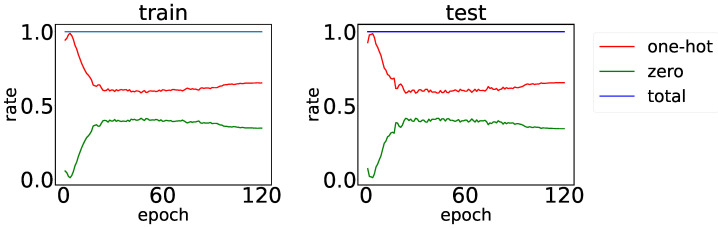
The averaged percentage of train and test set subcodes reaching one-hot or zero vectors. The total percentage rises to nearly 1 at the early stage of the training.

**Figure 5 sensors-23-02820-f005:**
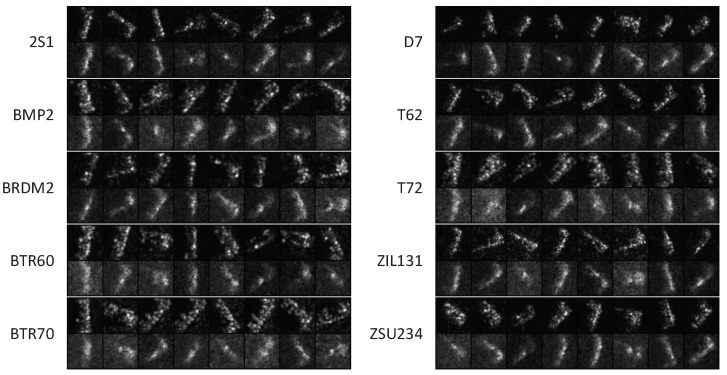
The prototype visualization results (bottom) and their most similar training samples (top). Some visualizations significantly differ from the most similar samples.

**Table 1 sensors-23-02820-t001:** The number of MSTAR train and test set samples.

Class	Type	Train Num.	Test Num.
2S1	b01	299	274
BMP2	9563	233	195
BRDM2	E-71	298	274
BTR60	k10yt7532	256	195
BTR70	C71	233	196
D7	92v13015	299	274
T62	A51	299	273
T72	132	232	196
ZIL131	E12	299	274
ZSU234	d08	299	274

**Table 2 sensors-23-02820-t002:** The grid search results of code segmentation number *M*, SNN iteration *I*, and two dropout rates $D_{conv}$ and $D_{class}$. SNN is ablated when *I* is zero, and the dropout layer is ablated when the corresponding *D* is zero. Results, ordered from largest to smallest, are labeled on a red–white–blue scale.

(a) $D_{\mathrm{conv}}=0.0$, $D_{\mathrm{class}}=0.0$						
	I	**0**	**8**	**16**	**32**	**64**
M						
32			96.99	97.66	97.63	98.26
64			96.80	97.02	98.29	96.50
128		97.51	96.51	96.99	96.67	96.94
256			98.11	98.13	96.70	98.21
512			97.53	97.39	97.41	97.87
(**b**) $D_{\mathrm{conv}}=0.1$, $D_{\mathrm{class}}=0.0$						
	I	**0**	**8**	**16**	**32**	**64**
M						
32			96.37	97.18	97.74	98.18
64			98.70	96.68	97.50	97.69
128		97.66	96.99	97.09	96.78	98.35
256			97.33	98.44	97.99	98.09
512			98.08	97.67	96.69	97.61
(**c**) $D_{\mathrm{conv}}=0.0$, $D_{\mathrm{class}}=0.75$						
	I	**0**	**8**	**16**	**32**	**64**
M						
32			97.63	98.45	97.20	98.27
64			97.94	97.68	98.57	97.15
128		97.80	98.20	98.33	97.52	98.31
256			98.49	97.94	97.10	98.02
512			97.34	98.06	97.81	98.11
(**d**) $D_{\mathrm{conv}}=0.1$, $D_{\mathrm{class}}=0.75$						
	I	**0**	**8**	**16**	**32**	**64**
M						
32			98.35	98.47	98.15	98.43
64			98.72	97.98	98.60	98.43
128		98.00	98.43	98.52	98.60	98.64
256			98.52	98.93	98.52	98.27
512			98.35	98.15	98.56	98.64

**Table 3 sensors-23-02820-t003:** Recognition results comparison. The results in the upper part of the table do not use data manipulation methods, and the lower methods are labeled with specific approaches.

Method	Acc. (%)
A-ConvNet [36]	96.49
2-VDCNN [37]	97.81
3-VDCNN [37]	98.17
Pruned-70 [38]	98.39
CCAE [13]	98.59
MSNN (proposed)	**98.93** ^1^
A-ConvNet with data augmentation [36]	99.13
DeepMemory with data augmentation [39]	99.71
LADL with data denoising [40]	**99.99** ^1^

^1^ Optimal result.

**Table 4 sensors-23-02820-t004:** The confusion matrix of the recognition results.

Class	2S1	BMP2	BRD M2	BTR 60	BTR 70	D7	T62	T72	ZIL 131	ZSU 234	Acc. (%)
2S1	263	0	5	0	0	0	5	0	1	0	95.99
BMP2	0	196	0	0	0	0	0	0	0	0	100
BRDM2	2	1	266	1	2	0	0	0	4	0	97.08
BTR60	0	0	4	191	0	0	0	0	0	0	97.95
BTR70	0	0	0	0	196	0	0	0	0	0	100
D7	0	1	1	0	0	272	0	0	0	0	99.27
T62	0	0	0	0	0	0	273	0	0	0	100
T72	0	0	0	0	1	0	0	195	0	0	99.49
ZIL131	0	0	0	0	0	0	0	0	274	0	100
ZSU234	0	0	0	0	0	0	0	0	0	274	100
**Total (%)**	98.93										

**Table 5 sensors-23-02820-t005:** The training time of different networks in one epoch.

Method	Time (s)
LADL	2.7
A-ConvNet	3.1
CCAE	4.3
MSNN (proposed)	4.6
DeepMemory	9.1

## Data Availability

Publicly available datasets were analyzed in this study. This data can be found here: https://www.sdms.afrl.af.mil/index.php?collection=mstar, accessed on 1 March 2023.

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
