# Peer review of "Modern Synergetic Neural Network for Synthetic Aperture Radar Target Recognition"

_sensors, 2023, doi:10.3390/s23052820_

Round 1

Reviewer 1 Report

In this paper, the authors proposed an improved Synergetic Neural Network model named Modern Synergetic Neural Network (MSNN) to transform the feature extraction process into the prototype self-learning process by the deep fusion of autoencoder and Synergetic Neural Network

Here are the comments to the authors, with due respect to their efforts in identifying the problem, coming up with a solution, experimenting, and bringing this article to the present format:

1. Title is clear and indicates the work done

2. Abstract is clear. The need and motivation behind the work are explained. The authors need to abbreviate SNN in the abstract.

3. Keywords are representing the work done. Maybe adding more appropriate keywords increase the visibility of this work for interested readers. Remove the name of the data set from the keywords

4. Introduction and Background sections are well written. In the introduction, section authors must add the following content: Objectives of this work, Novel contributions, Target users or beneficiaries of this work, the structure of the paper

5. Proposed model is explained clearly

6. Experimental section consists of dataset description, and experimental configuration. However, the authors need to explain the following points:

- reasons for opting for the parameters specified in the MSNN setup with supporting experimental evidence

- must present and describe more experimental results (by varying hyper-parameters)

- The reasons for the better performance of A-ConvNet with data augmentation

- When another model is performing better, what is the purpose of this model?

- In table-2, update the last row as MSNN (Proposed) from MSNN (ours)

- In table-3, the accuracy of the 2S1 object is low (95.99). What are the reasons behind that? What are the other similar targets to 2S1 causing the reduction in accuracy (misclassification)? How to overcome that problem? 

7. Conclusions are clear and implicative. However, the authors must explain the limitations of the proposed method, unexplored perspectives, future directions, etc.

References section:

- There are 38 references in total. The number of papers referred to and cited from the last five years is as follows:

2023 - 0 (Preprints/Accepted articles/Early access articles)

2022 - 0

2021 - 0

2020 - 4

2019 - 5

- The authors must refer to similar works published in recent years. If there are no similar works in the literature, the authors must discuss the reason for that and also present the difference between the work done in the published articles and the work done in this paper

- For Reference 35, the Journal name is missing

Reviewer 2 Report

This paper presents a Modern Synergetic Neural Network (MSNN) which transforms the feature extraction process into the prototype self-learning process by the deep fusion of autoencoder (AE) and SNN. Experimental studies have demonstrated that the proposed MSNN can improve the learning efficiency and performance stability using the real-life SAR dataset.

Pros: +++

1. The proposed MSNN has some interesting novelties.

2. Experiments on real SAR dataset validate the performance improvement. 

Cons: ---

1. The ablation study is missing, given that the proposed MSNN architecture has many parameters that can be configured. The authors are encouraged to conduct more experiments to show the performance stability.

2. The computational complexity analysis is missing, and the authors are also encouraged to add some discussions and comparisons with other methods.

Overall, this paper is well written and organized. But there are still some issues existing in this paper, which should be addressed in their future revision. 

Round 2

Reviewer 1 Report

As a reviewer, I am pleased to report that the revisions made to the manuscript have addressed the concerns raised in my previous review. The updates have improved the clarity and depth of the research. I believe the paper is now ready for acceptance and look forward to seeing it published. Thank you to the authors for their diligent efforts in incorporating the suggestions. 

Author Response

Thank you again for your recognition of our research. We can feel the diligence in your review, and your rigorous attitude is an example for us to follow.

As for the English language and style, we have invited several native speakers to proofread the text. Their comments and suggestions allowed us to embellish the article further. We hope that the manuscript will be more fluent and meet your standards.

We believe our responses have well addressed your concerns, and we hope our revised manuscript can be accepted for publication. Thank you very much for your time. We are looking forward to hearing from you.